# Postoperative Outcomes and Reinterventions Following Fenestrated/Branched Endovascular Aortic Repair in Post-Dissection and Complex Degenerative Abdominal and Thoraco-Abdominal Aortic Aneurysms

**DOI:** 10.3390/jcm11164768

**Published:** 2022-08-16

**Authors:** Bright Benfor, Julia Högl, Ryan Gouveia e Melo, Jan Stana, Carlota Fernandez Prendes, Maximilian Pichlmaier, Barbara Rantner, Nikolaos Tsilimparis

**Affiliations:** 1Department of Vascular Surgery, Ludwig Maximilian University Hospital, 81377 Munich, Germany; 2Department of Heart Surgery, Ludwig Maximilian University Hospital, 81377 Munich, Germany

**Keywords:** fenestrated endovascular aortic repair, branched endovascular aortic repair, thoracoabdominal aortic aneurysms, complex aortic aneurysms, FEVAR, BEVAR

## Abstract

Background: The outcome of FBEVAR in post-dissection thoracoabdominal aortic aneurysms has not been well established in the literature. The aim of this study was to compare midterm outcomes following FBEVAR in post-dissection aneurysms to degenerative aneurysms. (2) Methods: This was a retrospective review of all patients undergoing FBEVAR in a single center between 2017 and 2020. The baseline characteristics, intraoperative details, and postoperative outcomes of patients with post-dissection aneurysms were compared to those with degenerative outcomes. The primary end point was unplanned reinterventions. Cox regression analysis was performed to identify the determinants of worse outcomes. Results: A total of 137 subjects with a mean age of 70 ± 10 years were included in the study, out of which 30 presented post-dissection aneurysms (22%). Custom-made devices were employed in 119 patients, off-the-shelf devices in 13 and physician-modified endografts in 5, with incorporation in 505 target vessels. The technical success rate was comparable in both groups (97% vs. 98%, *p* = 0.21). However, the one-year freedom from unplanned reintervention was lower in the post-dissection group (67% vs. 89%, *p* = 0.011). Conclusion: FBEVAR in post-dissection aneurysms is associated with a favorable technical success rate, but reintervention rates remain high. Long procedural duration and the use of adjunctive techniques are associated with increased risk of reinterventions.

## 1. Introduction

Aneurysmal degeneration is a frequent chronic complication of aortic dissection with an estimated 5-year incidence of 73% among patients with type B dissection on medical therapy alone versus 63% after thoracic endovascular aortic repair [1]. As such, aortic dissection is the second most common etiology of aortic aneurysms after atherosclerotic degenerative aneurysms, accounting for 15–20% of all thoracoabdominal aortic aneurysms (TAAAs) [2]. The risk of rupture and of aneurysmal-related death in TAAAs is estimated at 3.7% and 12% per year, respectively, with aneurysmal size being the major predictive factor of rupture. Thus, timely intervention to exclude large aneurysms plays a critical role in the survival of these patients. Endovascular aortic repair using fenestrated and branched endografts to incorporate aortic branch vessels offers a less invasive approach to aneurysmal exclusion and may potentially reduce the rate of perioperative morbidity and mortality [3]. Previous studies have reported promising results with fenestrated and branched endovascular aortic repair (FBEVAR) in the treatment of complex aortic aneurysms [3,4,5]. However, there is a scarcity of literature regarding the comparative outcomes of FBEVAR in post-dissecting aneurysms, and the influence of chronic aortic dissection on FBEVAR has not been well established. Their outcomes may differ from those of degenerative aneurysms due to anatomic complexity and intraoperative challenges. The aim of this present study was to compare the clinical characteristics and outcomes of fenestrated/branched endovascular aortic repair in post-dissection aneurysms to degenerative aneurysms, in an “all-comers” cohort.

## 2. Materials and Methods

### 2.1. Type of Study and Setting

This was a retrospective cohort study performed in the Vascular Surgery department of a tertiary referral center in Germany to evaluate the outcomes of fenestrated and branched endovascular aortic repair in the treatment of complex abdominal and thoraco-abdominal aneurysms, with a specific focus on post-dissection aneurysms. The study was approved by the Institutional Review Board, and all patients gave an informed consent prior to surgery.

### 2.2. Patient Population

Consecutive patients receiving complex endovascular aortic repair with either fenestrated or branched aortic endografts for degenerative or post-dissection abdominal and thoracoabdominal aneurysms between January 2017 and January 2020 were included in this study. Patients’ preoperative, surgical and follow-up data, which had been prospectively collected in the institution’s electronic medical records platform, were retrospectively analyzed. Data collected included demographics, cardiovascular risk factors, comorbidities, clinical characteristics, radioanatomical features, operative details and postoperative outcomes.

### 2.3. Preoperative Management and Planning

Prior to surgery, a computerized tomographic angiography (CTA) with slice thickness not greater than 1.25 mm was obtained of the entire aorta and iliofemoral vessels in all patients for preoperative planning and sizing of endograft. Thoracoabdominal aneurysms were described according to their extent based on the Crawford classification (i.e., Crawford extents I-IV), and abdominal aneurysms were classified as paravisceral, pararenal, juxtarenal or infrarenal based on the SVS Fenestrated and Branched EVAR Reporting standards’ proposed anatomical site classification of TAAAs and AAAs [6]. Furthermore, Crawford extents I-III were termed as being extensive, whereas Crawford extent IV and AAAs were described as being non-extensive. Indication for treatment included aneurysm size ≥55 mm or evidence of progressive growth ≥10 mm per year or 5 mm per 6-month period. Symptomatic and ruptured aneurysms were treated irrespective of aneurysmal size. Further preoperative workup was performed to assess patients cardiac, respiratory, renal and other vital parameters before surgery. The physical status of patients was evaluated using the American Society of Anesthesiology (ASA) preoperative score. Preoperative medical management included blood pressure medication, antiplatelet therapy, anti-diabetic treatment, optimization of cardiac medication and smoking cessation. Patients were either treated using custom-made devices or an off-the-shelf thoracoabdominal multibranched endograft (T-branch^®^, Cook medical, Bloomington, IN, USA). Surgeon-modified fenestrated-branched endografts were also used in urgent situations when a T-branch^®^ was not anatomically feasible. All proximal components were oversized to 10–20% of the proximal landing zone diameter as determined on preoperative imaging. Likewise, iliac limbs were oversized to 10–15% of the distal seal zone diameter. Generally, caudally-oriented directional branches were preferred to fenestrations in patients with wide visceral aorta (≥30 mm). Inner side branches as well as retrograde branches were selectively used to incorporate target vessels in some patients, depending on the anatomy and at the discretion of the surgeon.

### 2.4. Surgical Procedure and Follow-Up

Procedures were typically performed in a hybrid room under general anesthesia. Spinal drainage was used in patients with extensive thoraco-abdominal aneurysms to reduce the risk of spinal cord ischemia (SCI). The drain was placed before the procedure and left in place for 48 to 72 h postoperatively. Similarly, Minimally Invasive Segmental Coil Embolization (MISACE) was performed in certain patients prior to the index procedure. Preoperative debranching of the left subclavian artery was required to extend the PLZ in certain patients with extensive TAAAs. Femoral access was achieved either by surgical cut-down in the groins or most commonly by percutaneous ultrasound-guided access with the use of suture-based preclosure devices (PROGLIDE, Abbott Vascular, Redwood City, CA, USA). When anatomically favorable, the right groin access was preferred for introduction of the main aortic components of the endograft, and fenestrations were generally cannulated from the contralateral femoral access. On the other hand, anterograde access from the arm or axillary region was generally used to cannulate caudally-oriented directional branches. However, in selected cases, transfemoral access was used to cannulate these branches. Endografts with precannulated fenestrations were employed in patients with extremely narrow or diseased contralateral iliac access. The choice of bridging stent type was at the discretion of the principal operator, with the choice of diameter and length being guided by preoperative imaging with an oversizing of 0–10%. All bridging stents were flared inside the aortic lumen to a diameter of +2 mm of the nominal diameter. Intraoperative fusion imaging guidance was routinely performed in our cohort, and postoperative CTA was required in all patients before discharge. After the procedure, patients were closely monitored in a dedicated intensive care unit (ICU) or intermediate care (IMC) for 24 to 48 h before being transferred to a normal ward. Upon discharge and in the absence of any complications, patients were followed up at 30 days with a CTA, then at 6 months with a duplex ultrasound, at 12 months with CTA and yearly thereafter with CTA.

### 2.5. Outcome Measures and Definition of Variables

The primary end point of our study was unplanned graft-related reintervention defined according to the SVS reporting standards on FBEVAR [6] as any unplanned secondary procedure designed to treat the underlying aortic disease, a complication of the main aortic endograft or to treat a branch vessel instability. This consisted of open conversion, interventions for endoleaks, treatment of target vessel stenosis, thrombosis and graft limb occlusions. Surgical or endovascular management of access complications and non-graft limb-related acute limb ischemia was not included in this definition. Secondary end points were mortality, cumulative rate of types I and III endoleaks, aneurysmal rupture, aneurysm sac progression or shrinkage, target vessel instability and major adverse events including access site complications, acute limb ischemia, spinal cord ischemia, cardiac dysfunction, kidney impairment, mesenteric ischemia, respiratory dysfunction necessitating mechanical ventilation or prolonged intubation, postoperative stroke and sepsis. Sac progression was defined by aneurysm growth ≥5 mm, and shrinkage was defined as a decrease in aneurysm size ≥5 mm compared to baseline diameter. Type 1b endoleaks which were present on completion of the angiogram and intentionally left for a second staged iliac side branch or distal extension were not considered adverse events, nor were the corresponding secondary procedures considered as reinterventions in this study. The primary exposure studied was the presence of chronic dissection in the aneurysmal segment being treated.

### 2.6. Statistical Analysis

Continuous variables were expressed either as mean ± standard deviation or as median [range] depending on their distribution, while categorical variables were expressed as percentages. Patients were divided into two groups depending on the etiology of their aneurysm (i.e., degenerative or post-dissection) and compared for differences in demographics, anatomical features, clinical presentation, intraoperative and postoperative outcomes. Categorical variables were compared using a chi-squared test or a Fisher exact test where applicable, and continuous variables were compared using a two-sample t-test when normally distributed or a Wilcoxon Rank Sum test where variables were not normally distributed. Kaplan–Meier survival curves and failure plots were used to describe and compare freedoms from reintervention, mortality and types I and III endoleaks and mortality. All statistics were two-tailed, and a *p*-value < 0.05 was considered significant. Statistical analyses were performed with IBM SPSS Statistics for Windows, Version 26.0. (IBM Corp., Armonk, NY, USA).

## 3. Results

### 3.1. Demographics and Baseline Characteristics

A total of 137 patients underwent fenestrated or branched endovascular aortic repair during the study period. Out of these, 107 (78%) presented degenerative aneurysms and 30 (22%) presented post-dissection aneurysms. Their mean age was 70 ± 10 years and 31 (23%) were female. Patients with post-dissection aneurysms were significantly younger than those presenting degenerative aneurysms, with a mean age of 60 ± 11 years versus 73 ± 7 years, respectively (*p*-value < 0.0001), while patients with degenerative aneurysms tended to be active smokers (39.2% vs. 10.7%, *p*-value < 0.005) and trended towards more cardiovascular comorbidities than their post-dissection counterparts (Table 1). On the other hand, there was no significant difference between both groups in ASA physical score, cardiac ejection fraction or baseline glomerular filtration rate (GFR). Degenerative aneurysms were less extensive compared to post-dissection aneurysms (19.6% vs. 70.0%; *p*-value < 0.0001) and MISACE, and prior aortic repair was more frequently encountered in the post-dissection group (Table 1). In addition, significant access vessel calcification was less common in post-dissection aneurysms (13.3% vs. 38.3% *p*-value = 0.010)

### 3.2. Procedural Details

Out of the 137 procedures, 117 were elective, 14 were emergency procedures to treat ruptured aneurysms, and 6 were urgent repairs of symptomatic intact aneurysms. Urgent/emergency repair was more frequent in the degenerative aneurysm group (21.5% vs. 3.3%; *p* = 0.021). Custom-made devices were employed in 119 patients, off-the-shelf multibranched devices (T-branch^®^, Cook Medical) in 13 patients, and surgeon-modified endografts in 5 cases, with femoral access being percutaneous in 123 patients (89.8%). A total of 505 target vessels were incorporated by 333 fenestrations (65.9%), 150 directional branches (29.7%), and 22 scallops (4.4%). There was no significant difference between the two groups regarding the average number of target vessels incorporated (3.9 ± 0.3 vs. 3.6 ± 0.8, *p*-value = 0.095); however, post-dissection aneurysms were more likely to have coeliac trunk incorporation (96.7% vs. 72%, *p*-value = 0.004). The modality of target vessel incorporation and types of bridging stents used are illustrated in Figure 1 and Figure 2, respectively. Accessory renal arteries were incorporated in a total of seven subjects (5%). Adjunctive procedures performed included aortic bifurcation relining in 14 cases (10.2%) and iliac side branch implantation in 31 cases (22.6%). Post-dissection aneurysms presented a higher tendency for adjunct procedures (56.7% vs. 22.4%, *p*-value < 0.0001). In addition, the volume of contrast product used was significantly higher in post-dissection aneurysms (median of 400 mL vs. 300 mL, *p*-value = 0.009). However, there was no significant difference in fluoroscopy time nor in total procedure duration. Table 2 presents a comprehensive report of procedural details.

### 3.3. Early Postoperative Outcomes

Technical success was achieved in 97% of the entire cohort (133 patients) without significant difference between groups. The median ICU and total postoperative length of stay (L.O.S) for the entire cohort were 4 (1–4) and 9 (7–16) days respectively, with no significant difference between groups. Furthermore, there was no significant difference between the two groups in terms of major adverse cardiovascular events (MACE) although degenerative aneurysms trended towards higher rates of MACE (14% vs. 3.5%; *p* = 0.19). Postoperative spinal cord injury was diagnosed in 10 degenerative aneurysms (9.4%) versus 1 post-dissection aneurysm (3.3%; *p*-value = 0.45), with paraplegia occurring in a total of 5 patients (3.8%). No type 1a endoleak was recorded at 30 days but type 1b was encountered in a total of seven patients (4.7%) and was significantly higher in post-dissection aneurysms (20% vs. 0.9%). On the other hand, the rate of type II endoleaks was similar in both groups (7.5% versus 6.7%; *p*: NS). A total of four deaths (2.9%) occurred at 30 days, all four being degenerative aneurysms. The 30-day outcomes are detailed on Table 3.

### 3.4. Short-Term Outcomes

During a mean follow-up of 15 months, no type Ia endoleak was recorded, type Ib was present in 8/133 patients and type Ic in 3 (2.3%). There was no significant difference in the occurrence of type II endoleak between both groups (Table 4); however, the Kaplan–Meier failure estimate of the one-year cumulative rate of types I and III endoleaks was higher in post-dissection aneurysms (35% vs. 13%, *p*-value = 0.013) as shown on Figure 3. Target-vessel-related interventions (20% vs. 4.9%: *p*-value = 0.016) and sac progression (36.0% vs. 15.9%, *p*-value = 0.029) were also more frequent in post-dissection aneurysms. However, no aneurysmal rupture was recorded in any of the groups. The Kaplan–Meier survival estimate of 1-year freedom from graft-related reintervention is illustrated on Figure 4. It was 89 ± 4% for degenerative aneurysms versus 67 ± 10% for post-dissection aneurysms (*p*-value = 0.011). The factors identified to be associated with unplanned reinterventions in univariate analysis include post-dissection aneurysm, age < 60 years, total procedural time > 240 min and realization of adjunctive procedures (Table 5). A total of nine deaths occurred during follow-up, with an estimated one-year freedom from all-cause mortality of 100% in the post-dissection group versus 91 ± 3% in degenerative aneurysms (Figure 5).

## 4. Discussion

In this study, we compared the clinical characteristics and outcomes of FBEVAR in post-dissection aneurysms to degenerative aneurysms and found the former to be associated with higher rates of types I and III endoleaks and target vessel instability. Moreover, post-dissection aneurysms had a tendency for higher rates of unplanned reinterventions. Unlike degenerative aneurysms, the safety and effectiveness of FBEVAR in post-dissection aortic aneurysms have not been sufficiently established in the literature. However, recent studies have reported favorable outcomes and shown that FBEVAR is feasible in this very important sub-group of patients [7,8,9]. Oikonomou and colleagues [9] previously reported a technical success rate of 96% and a perioperative mortality of 5.6% in 71 patients undergoing FBEVAR for post-dissection aneurysms. Similarly, Spear et al. [10] reported a technical success rate of 93% with a 30-day mortality of 4.7%, while Tenorio and colleagues [8] reported similar high technical success rates between post-dissection aneurysms and degenerative aneurysms (100% vs. 98.8%; *p*-value = 0.14). With a technical success rate of 97% in post-dissection aneurysms, our cohort compares well with previous studies and, similar to Tenorio [8], did not report any significant difference between the two groups. Despite satisfactory technical success rates, unplanned reinterventions continue to be a major issue in FBEVAR and more importantly, in post-dissection aneurysms [11,12,13,14]. Oikonomou reported a cumulative reintervention rate of 20% at 1 year, the majority of which were due to target vessel instability and endoleak [9]. Similarly, Marques de Maniro [7] and colleagues reported a 25% reintervention rate in their cohort of 55 patients with post-dissection aneurysms, while Yuk et al. documented a reintervention rate of 30% with all of them occurring within less than 6 months of index procedure [15]. Furthermore, in comparing post-dissections to degenerative aneurysms, Tenorio [8] reported higher rates of reinterventions in the former (40% vs. 30%). These consistent high rates of reinterventions reported in the literature are a clear indicator of the anatomical complexity and the consequential technical challenges presented by post-dissection aneurysms [8]. As showcased in our cohort, post-dissection aneurysms are mostly extensive TAAA which may require more proximal sealing zones as well as incorporation of all four reno-visceral arteries to achieve successful aneurysmal exclusion [16]. Moreover, the frequent extension of disease into the iliac arteries could account for the high rate of type Ib endoleak observed despite adequate oversizing of the stent–graft. In addition, the presence of intimal flap and thrombosed false lumen in target vessels make their catheterization and expansion of bridging stent more difficult compared to degenerative aneurysms and may lead to longer operating times as well as the need to resort to adjunctive techniques and additional stent–graft components to achieve satisfactory target vessel incorporation [17]. It is well established that the higher the number of stent–graft components, the greater the potential for failure of attachment sites leading to endoleaks and the need for reinterventions [18]. Other major causes of unplanned reinterventions in FBEVAR are graft-limb occlusions and target vessel instability [11,12,19,20,21]. This was corroborated by our study, with graft limb occlusions occurring in 5% of the cohort, while target vessel relining and recanalization procedures were performed in 11/27 reinterventions. Certain authors have suggested the use of preloaded catheters to facilitate the cannulation of target vessels in post-dissection aneurysms [8,22] as well as the use of cone beam CT imaging for early detection and assessment of bridging stent compression by false lumen [23,24]. Other bailout techniques for difficult target vessel catheterization described in the literature include the balloon-anchoring technique [25], the loop technique [26], the snare-ride [27] and the use of steerable sheaths for retrograde access to antegrade branches [28]. Furthermore, authors such as Oikonomou suggest that using longer bridging stents to achieve adequate sealing in target vessels may help reduce reintervention rates [9]. Regarding bridging stents, it is noteworthy to mention that there are currently no dedicated stents for FBEVAR, and there are no clear guidelines or standardized practice in the choice of bridging stents, which mostly depends on the surgeon’s preferences. In a recent systematic review, Mezzetto and colleagues [29] reported that 40% of all bridging stents were balloon-expandable, and 28% were self-expandable, with the remaining 36% undefined. This situation may, however, change in the future as studies are currently underway to evaluate the safety and efficacy of various bridging stents in FBEVAR [30,31,32,33]. The higher rate of reinterventions occurring in the post-dissection group in our series could also be interpreted in the light of higher mortality rates in degenerative aneurysms. Owing to more frequent cardiac comorbidities in this latter group, it could be that some of them are actually dying before reintervention-worthy complications are detected. One particularity of our cohort is the relatively high number of patients receiving concurrent iliac side-branch device implants (23% overall). This proportion was significantly higher in the post-dissection group (43%) and should be considered when interpretating the results of this series, especially with regards to intra-procedural metrics such as total procedural times, radiation doses and amount of contrast volume used.

### Limitations

Limitations of this study include its retrospective nature which is inherently a source of information bias. In addition, contrary to some previous studies [8], our cohort encompassed all extents of aneurysms with unequal distributions between the two groups. This, coupled with other differences in baseline characteristics between the two groups, is a potential confounding factor for reintervention rates. Ideally, the risk of bias could be reduced by propensity matching; however, the relatively small number of post-dissection aneurysms in our cohort would not allow for a robust propensity study nor meaningful multivariate analysis. For this reason, we limited this study to univariate analysis and were thus unable to establish an independent association between the type of aneurysm treated and reintervention. In this regard, it is remarkable that Tenorio and colleagues [8] compared patients with similar aneurysmal extent and reported a trend towards higher reintervention rates in the post-dissection group, with borderline statistical significance (40% vs. 30%, *p*-value = 0.06). Furthermore, the proportion of patients presenting any type of endoleak was higher among post-dissection aneurysms in that study (76% vs. 43%, *p*-value = 0.001). These findings suggest a possible independent association, and future studies at a larger multicentric level are warranted to investigate this association further.

## 5. Conclusions

The outcomes of FBEVAR in post-dissection aneurysms are comparable with those of degenerative aneurysms in terms of technical success. They are also associated with lower perioperative mortality and excellent freedom from aneurysmal rupture. However, reintervention rates tend to be high in the short-term and are mostly due to endoleaks and target vessel instability. Propensity studies are warranted to further investigate the association between post-dissection aneurysms and unplanned graft-related reinterventions.

## Figures and Tables

**Figure 1 jcm-11-04768-f001:**
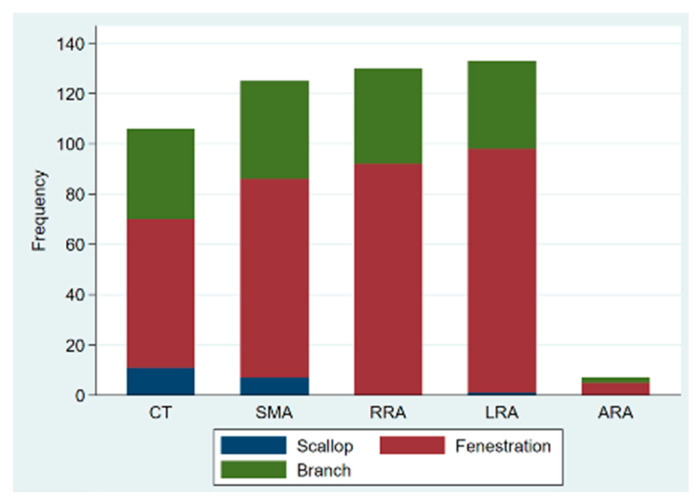
Bar graph of type of target vessel incorporation in fenestrated/branched endovascular aortic repair; CT: coeliac trunk; SMA: superior mesenteric artery; RRA: right renal artery; LRA: left renal artery; ARA: accessory renal artery.

**Figure 2 jcm-11-04768-f002:**
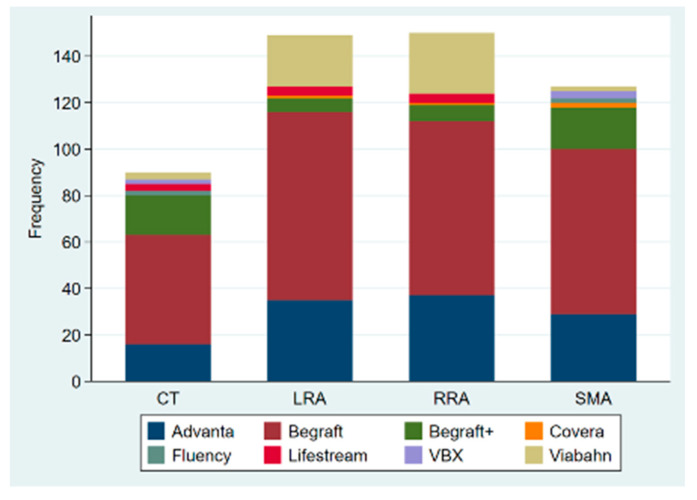
Bridging stents used to incorporate the reno-mesenteric arteries in fenestrated and branched endovascular repair of the aorta; CT: coeliac trunk; SMA: superior mesenteric artery; RRA: right renal artery; LRA: left renal artery.

**Figure 3 jcm-11-04768-f003:**
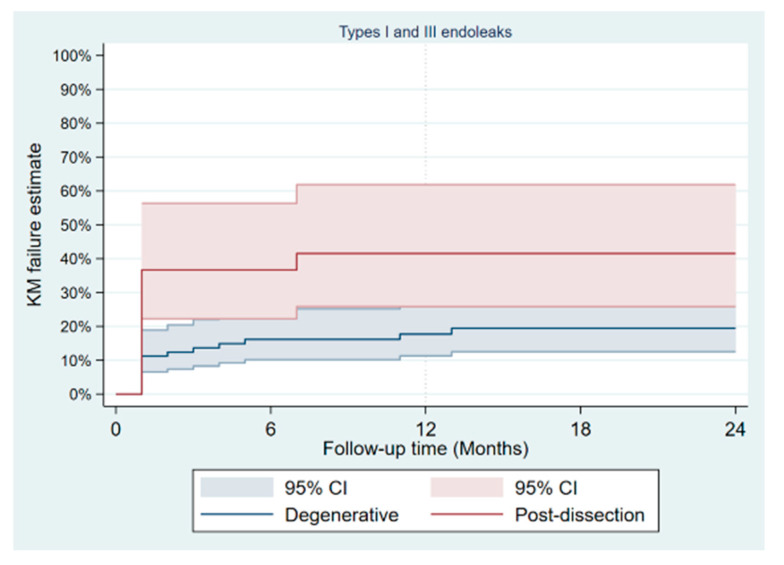
Kaplan–Meier failure plot of occurrence of type I and III endoleaks after fenestrated/branched endovascular aortic repair.

**Figure 4 jcm-11-04768-f004:**
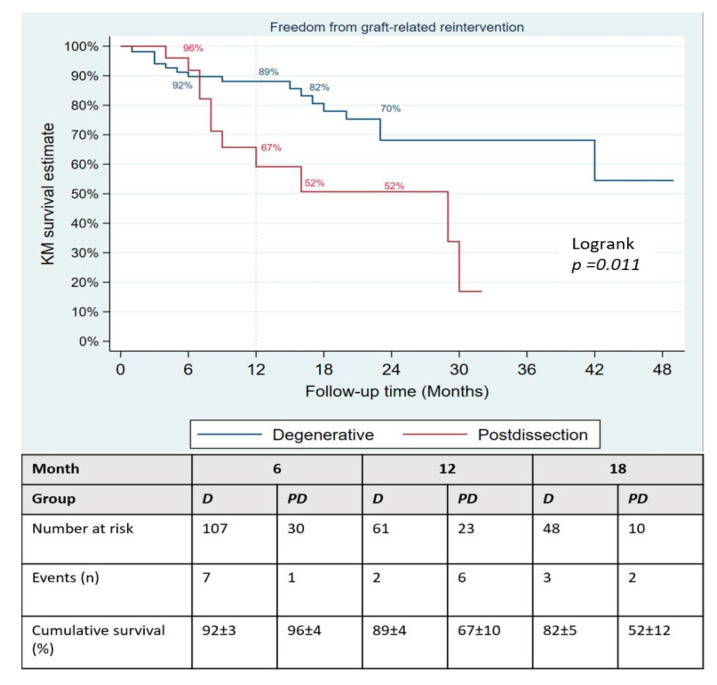
Kaplan–Meier failure plot of occurrence of type I and III endoleaks after fenestrated/branched endovascular aortic repair.

**Figure 5 jcm-11-04768-f005:**
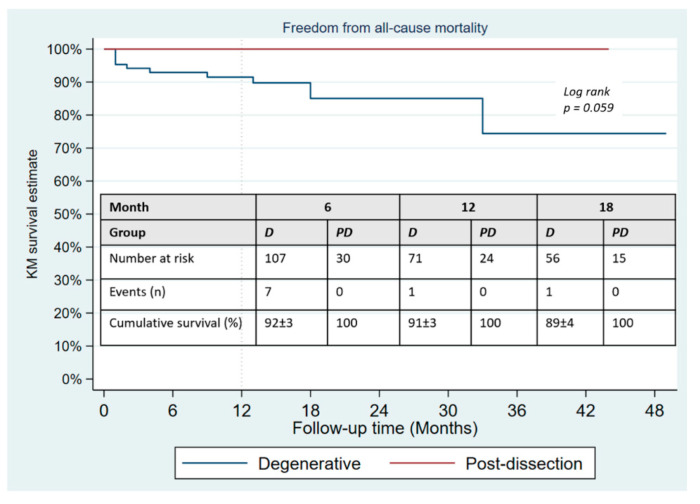
Kaplan–Meier survival estimates of freedom from all-cause mortality in patients undergoing fenestrated/branched endovascular aortic repair.

**Table 1 jcm-11-04768-t001:** Demographics and baseline characteristics of patients undergoing fenestrated and branched endovascular repair for thoraco-abdominal and abdominal aneurysms.

Variable *	All Patients (*n* = 137)	Degenerative Aneurysms (*n* = 107)	Post-Dissection Aneurysms (*n* = 30)	*p*-Value
**Age, years**	**70.4 ± 9.9**	**73.3 ± 7.3**	**59.9 ± 10.9**	**0.000**
Gender, Female	31(22.6)	21 (19.6)	10 (33.3)	0.113
BMI, kg/m^2^	26.8 ± 4.9	26.6 ± 4.9	27.4 ± 4.9	0.416
ASA score				0.443
2	3 (2.2)	3 (2.8)	0 (0.0)	
3	101 (73.7)	80 (74.7)	21 (70.0)	
4	32 (23.4)	23 (21.5)	9 (30.0)	
Ejection fraction, %	62.7 ± 11.7	61.4 ± 12.6	66.9 ± 6.3	0.099
Cardiovascular Comorbidities				
**CAD**	**35 (25.5)**	**33 (30.8)**	**2 (6.7)**	**0.007**
MI	9 (6.6)	8 (7.5)	1(3.3)	0.683
Arrhythmia	35 (25.5)	29 (27.1)	6 (20.0)	0.416
CHF	17 (12.4)	16 (15.0)	1 (3.3)	0.086
Hypertension	112 (81.7)	85 (79.4)	27 (90.0)	0.213
Ischemic stroke or TIA	17 (12.4)	16 (15.0)	1 (3.3)	0.119
PAD	17 (12.4)	14 (13.1)	3 (10.0)	0.764
Diabetes mellitus	18 (13.1)	17 (15.9)	1(3.3)	0.122
**Active smoker**	**43 (31.4)**	**40 (37.4)**	**3 (10.0)**	**0.005**
COPD	24 (17.5)	21 (19.6)	3 (10.0)	0.283
GFR, ml/min	67.9 ± 23.0	67.0 ± 21.2	71.1 ± 28.3	0.381
Connective tissue disorder	2(1.5)	1 (0.9)	1 (3.3)	0.394
Antiplatelet therapy	120 (87.6)	94 (87.9)	26 (86.7)	0.999
Statin use	92 (67.2)	75 (70.1)	17 (56.7)	0.145
ACE inhibitors	95 (69.3)	74 (69.2)	21 (70.0)	0.984
Betablockers	92 (67.2)	68 (63.6)	24 (80.0)	0.101
Hostile abdomen	18 (13.2)	16 (15.0)	2 (6.7)	0.299
**Preoperative debranching**	**7 (5.1)**	**1 (0.9)**	**6 (20.0)**	**0.000**
**Preoperative MISACE**	**54 (39.4)**	**31 (29.0)**	**23 (76.7)**	**0.000**
**Prior aortic surgery**	**49 (35.7)**	**24 (22.4)**	**25 (83.3)**	**0.000**
Endovascular	28 (20.4)	18 (16.8)	10 (33.3)	
Open	10 (7.3)	4 (3.8)	6 (20.0)	
Both	11 (8.0)	2 (1.9)	9 (30.0)	
Aneurysm size	61.9± 15.4	63.2± 16.1	57.4± 11.9	0.067
Status of aneurysm				0.133
Asymptomatic	117 (85.4)	89 (83.2)	29(96.7)	
Symptomatic non-ruptured	14(10.2)	13(12.1)	1(3.3)	
Ruptured	6(4.4)	6(5.6)	0(0)	
**Extent of aneurysm**				**0.000**
Extensive TAAA (I-III)	42 (30.7)	21 (19.6)	21 (70.0)	
AAA/Crawford IV	95 (69.3)	86 (80.4)	9 (30.0)	
**Significant iliac calcification**	**45 (32.8)**	**41 (38.3)**	**4 (13.3)**	**0.010**
Iliac tortuosity index (*n* = 135)	1.3 ± 0.2	1.3 ± 0.2	1.3 ± 0.1	0.098

AAA: abdominal aortic aneurysm; ACE: angiotensin-converting enzyme; CAD: coronary heart disease; CHF: congestive heart failure; COPD: chronic obstructive pulmonary disease; MI: myocardial infarction; MISACE: minimally-invasive segmental artery coil embolization; PAD: peripheral heart disease; TAAA: thoracoabdominal aortic aneurysm. * Categorical variables are presented as *n*, (%) and continuous variables as mean ± standard deviation or median (inter-quartile range).

**Table 2 jcm-11-04768-t002:** Intraoperative details of patients undergoing fenestrated and branched endovascular thoraco-abdominal and abdominal aortic repair.

Variable *	All Patients	Degenerative Aneurysms	Post-Dissection Aneurysms	*p*-Value
	(*n* = 137)	(*n* = 107)	(*n* = 30)
Surgical setting				0.048
Elective	117 (85.4)	88 (82.2)	29 (96.7)	
Urgent/Emergency	20 (14.6)	19 (17.8)	1 (3.3)	
Type of endograft				0.052
Custom-made	111 (81.0)	83 (77.6)	28(93.3)	
Off-the-shelf	16 (11.7)	14 (13.1)	2 (6.7)	
Surgeon-modified	10 (7.3)	10 (9.3)	0(0.0)	
Inverted limb	17 (12.41)	16 (15.0)	1 (3.3)	0.088
Preloaded catheter	13 (9.5)	11 (10.3)	2 (6.7)	0.55
Percutaneous femoral access	123 (89.8)	96 (89.7)	27 (90.0)	0.99
N° of Target vessels	3.6 ± 0.8	3.6 ± 0.8	3.9 ± 0.3	0.095
CT	106 (77)	77 (72)	29(96.7)	0.004
SMA	125 (91)	95 (88.8)	30 (100)	0.068
RRA	130 (94.9)	103 (96)	27 (90)	0.177
LRA	134 (97.8)	107 (100)	30 (100)	0.999
ARA	7 (5)	7 (6)	0 (0)	0.347
N° of Target vessels ≥4	100 (73.0)	74 (69.0)	26 (86.7)	0.056
Type of incorporation				
n	505	388	117	
Scallop	22(4.4)	22(5.7)	0(0)	0.008
Fenestrations	333 (65.9)	241 (62)	92 (78.6)	0.001
Branches	150 (29.7)	125 (32)	25 (21.4)	0.024
Bridging stent diameter, mm				
CT	8 ± 2	8 ± 2	9 ± 1	0.008
SMA	8 ± 1	8 ± 1	8 ± 1	0.17
RRA	6 ± 1	6 ± 1	7 ± 1	0.040
LRA	7 ± 1	6 ± 1	7 ± 1	0.011
Bridging stent sealing length, mm ± SD				
CT	43 ± 26	45 ± 28	38 ± 18	0.26
SMA	46 ± 24	47 ± 23	45 ± 26	0.74
RRA	40 ± 24	40 ± 25	40 ± 23	0.97
LRA	40 ± 23	39 ± 23	42 ± 24	0.54
Adjunctive procedures	41 (29.9)	24 (22.4)	17 (56.7)	<0.001
Bifurcation relining	14 (10.2)	7 (6.5)	7 (23.3)	0.014
Iliac side branch	31 (22.6)	18 (16.8)	13(43.3)	0.002
Fusion	51 (37.3)	41 (38.3)	10 (33.3)	0.62
CSF drainage	69 (50.4)	45 (42.1)	24 (80.0)	<0.001
Contrast volume, mL	300 (200–400)	300 (200–400)	400 (300–500)	0.009
Fluoroscopy time, min	83 (61–115)	81 (60–116)	89 (71–115)	0.74
Radiation dose, mGy	3255 (2035–5889)	3134 (2035–5909)	4624 (1849–5736)	0.75
Total procedural time, min	261 (203–333)	255 (196–334)	272 (232–315)	0.73
Technical success	133 (97.1)	105 (98.1)	28 (97.1)	0.21
Endoleak at final angio	22 (16.0)	17 (15.9)	5 (16.7)	0.99
Type I	4 (2.9)	1 (0.9)	3 (10.0)	0.033
Type II	16 (11.7)	14 (13.1)	2 (6.7)	0.52
Type IIIc	2 (1.5)	2 (1.9)	0 (0.0)	0.45

CT: celiac trunk; LRA: left renal artery; RRA: right renal artery; SMA: superior mesenteric artery. *** Categorical variables are presented as *n*, (%) and continuous variables as mean ± standard deviation or median (inter-quartile range).

**Table 3 jcm-11-04768-t003:** Comparative analysis of 30-day outcomes of FBEVAR in post-dissection and degenerative aneurysms.

Outcome *	All (*n* = 137)		Degenerative Aneurysms *n* = 107		Post-Dissection Aneurysms *n* = 30		*p*-Value
MACE	16	(11.8)	15	(14)	1	(3.5)	0.19
Myocardial infarction	8	(5.8)	8	(7.5)	0	(0.0)	0.20
Acute heart failure	11	(8.0)	10	(9.9)	1	(3.3)	0.46
Ischemic stroke	4	(2.9)	4	(3.7)	0	(0.0)	0.57
Respiratory complications	10	(7.3)	9	(8.4)	1	(3.3)	0.69
Acute kidney injury	16	(11.7)	12	(11.2)	4	(13.0)	0.75
No dialysis	12	(8.8)	10	(9.4)	2	(6.7)	-
Temporary dialysis	3	(2.2)	1	(0.9)	2	(6.7)	-
Permanent dialysis	1	(0.7)	1	(0.9)	0	(0.0)	-
Spinal cord injury	10	(7.5)	9	(8.7)	1	(3.3)	0.45
Paraplegia	5	(3.8)	5	(4.9)	0	(0.0)	-
Paresis	5	(3.8)	4	(3.8)	1	(3.3)	-
Ischemic colitis	2	(1.5)	2	(1.9)	0	(0.0)	0.99
Access vessel complication							
Femoral access	10	(7.3)	8	(7.5)	2	(6.7)	0.999
Upper access	2	(1.5)	2	(1.90	0	(0.0)	0.999
Surgical wound complication	4	(2.9)	4	(3.7)	0	(0.0)	0.576
Endoleak							
Type I & III	21	(15.3)	11	(10.3)	10	(33.0)	0.004
Ia	0	(0.0)	0	(0.0)	0	(0.0)	-
Ib	7	(4.7)	1	(0.9)	6	(20.0)	<0.001
Ic	1	(0.7)	0	(0.0)	1	(0.8)	-
IIIa	6	(4.4)	6	(5.6)	0	(0.0)	0.33
IIIc	7	(5.1)	4	(3.7)	3	(10.0)	0.18
Type II	10	(7.3)	8	(7.5)	2	(6.7)	0.99
Postoperative LOS, days	9	(7–16)	9	(7–20)	8.5	(7–11)	0.50
ICU length of stay, days	2	(1–4)	2	(1–4)	3	(2–4)	0.89
Reintervention	8	(5.9)	7	(6.5)	1	(3.5)	0.99
Mortality	4	(2.9)	4	(3.7)	0	(0.0)	0.58

ICU: Intensive care unit; LOS: length of stay; MACE: major adverse cardiovascular event. * Categorical variables are presented as *n*, (%) and continuous variables as mean ± standard deviation or median (inter-quartile range).

**Table 4 jcm-11-04768-t004:** Short-term outcomes.

Outcome *	All (*n* = 133) ^†^		Degenerative Aneurysms *n* = 103 *		Post-Dissection Aneurysms *n* = 30		*p*-Value
Follow-up, months, m (sd)	15.2	(13.1)	15.2	(13.3)	15.3	(12.5)	0.82
Late Endoleak, *n* (%)	37	(27.8)	23	(22.3)	14	(46.7)	0.009
Type I	11	(8.8)	3	(2.8)	8	(30.0)	<0.001
Ia	0	(0.0)	0	(0.0)	0	(0.0)	-
Ib	8	(6.0)	3	(2.9)	5	(16.7)	0.015
Ic	3	(2.3)	0	(0.0)	3	(10.0)	0.011
Type II	11	(8.3)	8	(7.8)	3	(10.0)	0.71
Type III	15	(11.0)	12	(11.2)	3	(10.0)	0.85
IIIa	13	(9.5)	10	(9.4)	3	(10.0)	0.7
IIIb	0	(0.0)	0	(0.0)	0	(0.0)	-
IIIc	2	(1.5)	2	(1.9)	0	(0.0)	0.99
Graft limb occlusion	7	(5.3)	4	(3.9)	3	(10.0)	0.19
Aneurysm rupture	0	0	0	0	0	0	-
Aneurysm sac change, mm/year (*n* = 107)	−0.1	(−7.0; 3.0)	−0.95	(−7.0; 2.2)	1.6	(−2.8; 7.3)	0.40
Sac shrinkage	28	(26.2)	22	(26.8)	6	(26.2)	0.78
Sac progression	22	(20.6)	13	(15.9)	9	(36.0)	0.029
Unchanged diameter	57	(53.3)	47	(57.3)	10	(40.0)	0.13
Late reintervention	27	(20.3)	18	(17.5)	9	(20.3)	0.13
Target vessel relining/recanalization	11	(8.3)	5	(4.9)	6	(20.0)	0.016
CT	2	(1.5)	1	(1.0)	1	(3.3)	0.40
SMA	3	(2.3)	1	(1.0)	2	(6.7)	0.13
RRA	4	(3.0)	2	(1.9)	2	(6.7)	0.21
LRA	6	(4.5)	2	(1.9)	4	(13.3)	0.023
Death	9	(6.8)	9	(8.7)	0	0	0.094

CT: celiac trunk; LRA: left renal artery; RRA: right renal artery; SMA: superior mesenteric artery; * Categorical variables are presented as *n*, (%) and continuous variables as mean ± standard deviation or median (inter-quartile range). ^†^ Four deaths occurred at 30 days and are excluded from analysis.

**Table 5 jcm-11-04768-t005:** Univariate analysis of factors associated with graft-related reintervention in 137 patients undergoing fenestrated/branched aortic repair.

Variable	OR	CI_95_	*p*-Value
Total procedure time > 240 min	5.3	(1.7–16.2)	0.002
Adjunctive procedures	3.0	(1.3–7.1)	0.009
Age < 60 years	4.7	(1.7–13.0)	0.002
Female gender	1.9	(0.8–4.8)	0.177
Extensive TAAA	2.0	(0.5–4.5)	0.116
Post-dissection aneurysm	3.1	(1.2–7.6)	0.013

OR: odds-ratio; CI: confidence interval.

## Data Availability

The underlying data for this study will be made available upon reasonable request to the senior author, N.T.

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
