# Peer review of "Postoperative Outcomes and Reinterventions Following Fenestrated/Branched Endovascular Aortic Repair in Post-Dissection and Complex Degenerative Abdominal and Thoraco-Abdominal Aortic Aneurysms"

_jcm, 2022, doi:10.3390/jcm11164768_

Round 1

Reviewer 1 Report

Midterm results and reinterventions following fenestrated/branched endovascular aortic repair in post-dissection 3 and complex degenerative abdominal and thoracoabdominal  aortic aneurysms. 

1.     Main comments: This interesting and well-written paper is a retrospective review of all patients undergoing FBEVAR in a single centre between 2017 and 2020. The aim of the study was to compare midterm outcomes following FBEVAR in postdissection aneurysms to degenerative aneurysms.

2.     ** The limitations of the study are obvious and already recognized by the authors, the study is retrospective and monocentric, therefore likely to present biases. Moreover, as the post-dissection group is relatively small (n=30), it is impossible to carry out a valid multivariate study or a propensity study, which would have been desirable given the differences between the two groups including age, CAD, smoking habits, but also preoperative debranching, MISACE, prior aortic surgery, and extent of the aneurysm.

3.     ***Line 259: About multivariate analysis, the authors wrote, 'Upon multivariate analysis, postdissection 259 aneurysm was not significantly associated with unplanned reinterventions.' This multivariate analysis makes little sense, the numbers of the 'dissection' group (n=30) are too small to introduce into the calculation the 7 variables whose frequency is different between the two groups. It would be preferable to limit this study to univariate analyses with the calculation of confidence intervals.

4.     *** Table 5: It is obvious that there is a relationship between graft-related reinterventions and adjunctive procedures. Establishing a causal relationship between these two variables does not bring much and it is even inappropriate in a multivariate study since there is a known relationship between these variables.

5.     The reader can therefore ask the question whether it is these important differences, more than the pathology (degenerative aneurysm vs. post-dissection aneurysm) that are responsible for late reoperations. This point should be discussed as it makes the generalization of the results of the article questionable. In this area, a multicenter study would have been preferable.

6.     Lines 266-331: Discussion is well written and documented.

7.     Lines 331-337: Limitations of the study. This paragraph is incomplete and the limitations of the study are not limited to the extent of the aneurysm. Given the small number of the post-dissection group, the authors should be more critical, and, in my opinion, remove the multivariate analysis.

8.     In conclusion, this article deserves to be published if the authors recognize the limitations of the study and delete the multivariate analysis because of the limited numbers.

Author Response

  1. Main comments: This interesting and well-written paper is a retrospective review of all patients undergoing FBEVAR in a single centre between 2017 and 2020. The aim of the study was to compare midterm outcomes following FBEVAR in postdissection aneurysms to degenerative aneurysms.

Response: Thank you for your comments

  1. ** The limitations of the study are obvious and already recognized by the authors, the study is retrospective and monocentric, therefore likely to present biases. Moreover,as the post-dissection group is relatively small (n=30), it is impossible to carry out a valid multivariate study or a propensity study, which would have been desirable given the differences between the two groups including age, CAD, smoking habits, but also preoperative debranching, MISACE, prior aortic surgery, and extent of the aneurysm.

Response: We agree that a propensity matching would have been more meaningful for this study, but not feasible due to the limited number of postdissection aneurysms. We have acknowledged this in the discussion.

Changes: Page 13, lines 342-356: Also, contrary to some previous studies [8], our cohort encompassed all extents of aneurysms with unequal distributions between the two groups. This, coupled with other differences in baseline characteristics between the two groups, is a potential confounding factor for reintervention rates. Ideally, the risk of bias could be reduced by propensity matching, however the relatively small number of postdissection aneurysms in our cohort would not allow for a robust propensity study nor meaningful multivariate analysis. For this reason, we limited this study to univariate analysis, thus unable to establish an independent association between the type of aneurysm treated and reintervention. In this regard, it is remarkable that Tenorio and colleagues[8] compared patients with similar aneurysmal extent and reported a trend towards higher reintervention rates in the post dissection group, with borderline statistical significance (40% vs 30%, p-value =0.06). Furthermore, the proportion of patients presenting any type of endoleak was higher among postdissection aneurysms in that study (76% vs. 43%, p-value =0.001). These findings suggest a possible independent association and future propensity studies at a larger, multicentric level, are warranted to investigate this association further.

  1. ***Line 259: About multivariate analysis, the authors wrote,'Upon multivariate analysis, postdissection 259 aneurysm was not significantly associated with unplanned reinterventions.' This multivariate analysis makes little sense, the numbers of the 'dissection' group (n=30) are too small to introduce into the calculation the 7 variables whose frequency is different between the two groups. It would be preferable to limit this study to univariate analyses with the calculation of confidence intervals.

Response: We agree and have revised the manuscript accordingly. We have also acknowledged this limitation in the discussion.

Changes: Page 9, lines 250-254. The factors identified to be associated with unplanned reinterventions in univariate analysis include: Post-dissection aneurysm, age < 60 years, total procedural time > 240 minutes, and realization of adjunctive procedures (Table 5). 

  1. *** Table 5: It is obvious that there is a relationship between graft-related reinterventions and adjunctive procedures. Establishing a causal relationship between these two variables does not bring much and it is even inappropriate in a multivariate study since there is a known relationship between these variables.

Response: Agreed. We have limited the study to only univariate analysis

Changes:  Page 11, line 270-272. Please refer to table 5 in the revised manuscript

  1. The reader can therefore ask the question whether it is these important differences, more than the pathology (degenerative aneurysm vs. post-dissection aneurysm) that are responsible for late reoperations. This point should be discussed as it makes the generalization of the results of the article questionable. In this area, a multicenter study would have been preferable.

Response: It is difficult to tell without an appropriate propensity study. We believe that extent of aneurysm and differences in baseline characteristics aside, the specific anatomical features of dissection alone (i.e, presence of intimal flap, thrombosed false lumen, visceral artery take off from false lumen, etc) can contribute to the procedure being more challenging, and consequently more postoperative complications requiring reinterventions. These have been already discussed in the original manuscript. In a previous study by Tenorio and colleagues, the authors compared only patients with extensive aneurysms and reported a trend toward higher reintervention rates in the post dissection group, with borderline statistical significance (40% vs 30%, p-value =0.06). Furthermore, the proportion of patients presenting any type of endoleak was higher in this group (76% vs. 43%, p-value =0.001). These results suggest a possible independent association that merits to be investigated further in more robust studies

Changes: Page 13, lines 342-356. Also, contrary to some previous studies [8], our cohort encompassed all extents of aneurysms with unequal distributions between the two groups. This, coupled with other differences in baseline characteristics between the two groups, is a potential confounding factor for reintervention rates. Ideally, the risk of bias could be reduced by propensity matching, however the relatively small number of postdissection aneurysms in our cohort would not allow for a robust propensity study nor meaningful multivariate analysis. For this reason, we limited this study to univariate analysis, thus unable to establish an independent association between the type of aneurysm treated and reintervention. In this regard, it is remarkable that Tenorio and colleagues[8] compared patients with similar aneurysmal extent and reported a trend towards higher reintervention rates in the post dissection group, with borderline statistical significance (40% vs 30%, p-value =0.06). Furthermore, the proportion of patients presenting any type of endoleak was higher among postdissection aneurysms in that study (76% vs. 43%, p-value =0.001). These findings suggest a possible independent association and future propensity studies at a larger, multicentric level, are warranted to investigate this association further.

  1. Lines 266-331: Discussion is well written and documented.

Response: Thank you

  1. Lines 331-337: Limitations of the study. This paragraph is incomplete, and the limitations of the study are not limited to the extent of the aneurysm. Given the small number of the post-dissection group, the authors should be more critical, and, in my opinion, remove the multivariate analysis.

Response: Agreed. We have expounded further on the limitations of this study

Changes: Page 13, lines 341-356. Limitations of this study include its retrospective nature which is inherently a source of information bias. Also, contrary to some previous studies [8], our cohort encompassed all extents of aneurysms with unequal distributions between the two groups. This, coupled with other differences in baseline characteristics between the two groups, is a potential confounding factor for reintervention rates. Ideally, the risk of bias could be reduced by propensity matching, however, the relatively small number of postdissection aneurysms in our cohort would not allow for a robust propensity study or meaningful multivariate analysis. For this reason, we limited this study to univariate analysis, thus unable to establish an independent association between the type of aneurysm treated and reintervention. In this regard, it is remarkable that Tenorio and colleagues[8] compared patients with similar aneurysmal extent and reported a trend toward higher reintervention rates in the post dissection group, with borderline statistical significance (40% vs 30%, p-value =0.06). Furthermore, the proportion of patients presenting any type of endoleak was higher among postdissection aneurysms in that study (76% vs. 43%, p-value =0.001). These findings suggest a possible independent association and future propensity studies at a larger, multicentric level, are warranted to investigate this association further.

  1. In conclusion, this article deserves to be published if the authors recognize the limitations of the study and delete the multivariate analysis because of the limited numbers.

Response: Thanks for your kind consideration.

Reviewer 2 Report

Thank you for your submission on Journal of Clinical Medicine with the manuscript titled “Mid-term results and reinterventions following fenestrated/branched endovascular aortic repair in post-dissection and complex degenerative abdominal and thoraco-abdominal aortic aneurysms”. Some questions need to be answered.

Question 1

The authors mentioned “Mid-term results”, But the mean follow-up time was only 15 months (Page 9 line 229), and Kaplan Meier results were based on one-year estimate (in 3.4. Midterm outcomes section). The reviewer think the short-term results should be more suitable.

Question 2

It was confused that number of 2 groups of patients. There are 108 degenerative aneurysms and 29 post-dissection aneurysms in Table 1. However, there are 107 degenerative aneurysms and 30 post-dissection aneurysms in main text. Please check it.

Question 3

How much is the ratio of oversize of distal stentgraft? If there exist difference between 2 groups? Please explain the reason that the rate of type Ib endoleak is statistically higher in post-dissection aneurysms group.  

Question 4

The primary endpoint of the study was unplanned reinterventions. The results show that technical success rate was comparable in both groups (97% vs 98%, p =0.21) however the one-year freedom from unplanned reintervention was lower in the postdissection group (67% vs 89%, p=0.011). hence, the authors get the conclusion that FBEVAR in postdissection aneurysms is associated with favorable technical success rate but reintervention rates remain high. However, there were significant differences in the anatomical conditions of the 2  groups, such as Extensive TAAA (I-III) accounting for 70% in post-dissection aneurysms group and only 19.6% in degenerative aneurysms group. Is the conclusion scientific based on such inconsistent baseline cases? Please explain it.

Question 5

In Table 4, the rate of types I endoleaks was statistically higher in post-dissection aneurysms group but the rate of types III endoleak was no statistical difference between 2 groups. However, Kaplan Meier failure estimate of one-year cumulative rate of both types I and III endoleaks were higher in post-dissection aneurysms (35% vs 13%, p-value = 0.013) as shown on Figure 3. So, I hope the authors give a clear and concise explanation of mixing 2 type of endoleak.

Some other details need to be clarified or revised.

In Table 1, please note the full name of the important abbreviation below the table.

Cardiovascular Comorbidities  should be Cardiovascular Comorbidities, n(%) and too much lines have similar errors. Please check it carefully.

In Table 2, "%" is redundant for “125 (32%)” in line of Branches. Several N appeared in the column of All patients should be n.  Type of incorporation (n= 505)  should be Type of incorporation, n (%) and n= should be deleted in the next 3 columns.  Bridging stent diameter, mean ± SD should be Bridging stent diameter, mm, mean ± SD . Bridging stent sealing length, mean ± SD “ should be Bridging stent sealing length,mm, mean ± SD”. 

In page 5 line 187, missing punctuation before the word "however".

Author Response

Question 1

The authors mentioned “Mid-term results”, But the mean follow-up time was only 15 months (Page 9 line 229), and Kaplan Meier results were based on one-year estimate (in 3.4. Midterm outcomes section). The reviewer thinks the short-term results should be more suitable.

Response: Agree. We have revised the manuscript accordingly

Changes:

  1. Title change: Page 1 lines 1 -4: Postoperative outcomes and reinterventions following fenes-trated/branched endovascular aortic repair in post-dissection and complex degenerative abdominal and thoraco-abdominal aortic aneurysms.
  2. Page 9, line 239: 3.4. Short term outcomes

Question 2

It was confused that number of 2 groups of patients. There are 108 degenerative aneurysms and 29 post-dissection aneurysms in Table 1. However, there are 107 degenerative aneurysms and 30

post-dissection aneurysms in main text. Please check it.

Response: Thank you for pointing out this oversight. We confirm that the numbers are respectively 107 and 30, and have made the corresponding changes in Table 1

Changes: Page 4, line 174: Please see table 1 in the revised manuscript

Question 3

How much is the ratio of oversize of distal stentgraft? If there exist difference between 2 groups? Please explain the reason that the rate of type Ib endoleak is statistically higher in post-dissection aneurysms group.

Response: Stentgrafts were oversized to ~10% at the distal sealing zones in both degenerative and postdissecting aneurysms. We have clarified this in the methods. The high rate of type Ib endoleak can be attributed to more distal extension of disease into the iliac arteries in postdissection aneurysms. This point has been raised in the discussion

Changes:

  1. Page 3, lines 89-90. Likewise, iliac limbs were oversized to 10 -15% of the distal seal zone diameter
  2. Page 12 lines 302-304. Also, the frequent extension of disease into the iliac arteries could account for the high rate of type Ib endoleak observed, despite adequate oversizing of stent-graft.

Question 4

The primary endpoint of the study was unplanned reinterventions. The results show that technical success rate was comparable in both groups (97% vs 98%, p =0.21) however the one-year freedom

from unplanned reintervention was lower in the postdissection group (67% vs 89%, p=0.011). hence, the authors get the conclusion that FBEVAR in postdissection aneurysms is associated with favorable technical success rate but reintervention rates remain high. However, there were significant differences in the anatomical conditions of the 2 groups, such as ExtensiveTAAA (I-III) accounting for 70% in post-dissection aneurysms group and only 19.6% in degenerative aneurysms group. Is the

conclusion scientific based on such inconsistent baseline cases?

Please explain it.

Response:  Based on our data, we observed that reintervention rates tend to be high in postdissection aneurysms. This result corroborates that of previous studies which have been cited in the discussion. We do agree with you that we cannot conclude on an independent association given the significant baseline and anatomical differences between the two groups. Future propensity studies will be more instrumental in establishing an independent association between postdissection aneurysms and higher reintervention rates. We have acknowledged this limitation in the discussion and revised our conclusion.

Changes:

  1. Page 13, lines 342-356: Also, contrary to some previous studies [8], our cohort encompassed all extents of aneurysms with unequal distributions between the two groups. This, coupled with other differences in baseline characteristics between the two groups, is a potential confounding factor for reintervention rates. Ideally, the risk of bias could be reduced by propensity matching, however the relatively small number of postdissection aneurysms in our cohort would not allow for a robust propensity study or meaningful multivariate analysis. For this reason, we limited this study to univariate analysis, thus unable to establish an independent association between the type of aneurysm treated and reintervention. In this regard, it is remarkable that Tenorio and colleagues[8] compared patients with similar aneurysmal extent and reported a trend towards higher reintervention rates in the post dissection group, with borderline statistical significance (40% vs 30%, p-value =0.06). Furthermore, the proportion of patients presenting any type of endoleak was higher among postdissection aneurysms in that study (76% vs. 43%, p-value =0.001). These findings suggest a possible independent association and future propensity studies at a larger, multicentric level, are warranted to investigate this association further

  1. Page 13, lines 360-363. However, reintervention rates tend to be high in the short-term and are mostly due to endoleaks and target vessel instability. Propensity studies are warranted to further in-vestigate the association between postdissection aneurysms and unplanned graft-related reinterventions.

Question 5

In Table 4, the rate of types I endoleaks was statistically higher in post-dissection aneurysms group but the rate of types III endoleak was no statistical difference between 2 groups. However, Kaplan Meier failure estimate of one-year cumulative rate of both types Iand III endoleaks were higher in post-dissection aneurysms (35% vs 13%, p-value = 0.013) as shown on Figure 3. So, I hope the authors give a clear and concise explanation of mixing 2 type of endoleak.

Response: Type I and III endoleaks are considered as the most important endoleaks for assessing clinical success following EVAR, and usually lead to unplanned reinterventions, contrary to type II endoleaks which are only considered important  if they lead to sac expansion. Furthermore “Freedom from type I or III endoleak” has been recognized in the SVS reporting standards as a major longitudinal endpoint which needs to be reported in studies evaluating FBEVAR (Ref). We have clarified this in the methods.

Changes

Page 3, line 122-130. The primary endpoint of our study was unplanned graft-related reintervention defined according to the SVS reporting standards on FBEVAR[6] … Secondary end-points were mortality, cumulative rate of types I and III endoleaks,…

Some other details need to be clarified or revised. In Table 1, please note the full name of the important abbreviation below the table. “Cardiovascular Comorbidities “ should be “Cardiovascular

Comorbidities, n(%)” and too much lines have similar errors. Please check it carefully. In Table 2, "%" is redundant for “125 (32%)” in line of Branches. Several “N” appeared in the column of “All patients” should be “n”. “Type of incorporation (n= 505) “ should be “Type of incorporation, n (%) and “n=” should be deleted in the next 3 columns. “Bridging stent diameter, mean ± SD” should be “Bridging stent diameter, mm, mean ± SD “. “Bridging stent sealing length, mean ± SD “ should be “Bridging stent sealing length,mm, mean ± SD”. In page 5 line 187, missing punctuation before the word "however".

Response: Thank you for pointing out these errors.

Changes: We have specified at the bottom of each table that: Categorical variables are presented as n, (%) and continuous variables as mean ± standard deviation or median (inter-quartile range). All other corrections have been made in the revised manuscript.

Round 2

Reviewer 1 Report

The manuscript has been improved significantly. I have no other comment.

Reviewer 2 Report

All questions have been answered and the authors have revised the manuscript.  I am satisfied with these changes.